# Clinical efficacy and safety of interferon (Type I and Type III) therapy in patients with COVID-19: A systematic review and meta-analysis of randomized controlled trials

**Seungeun Ryoo**[1,2☯], **Dae-Hyup Koh**[1,3☯], **Su-Yeon Yu**[1,4], **Miyoung Choi**[1],
**Kyungmin Huh**[5], **Joon-Sup Yeom**[6], **Jung Yeon Heo**[7]*

**1** Division for Healthcare Technology Assessment Research, National Evidence-based Healthcare Collaborating Agency, Seoul, Korea, **2** Department of Public Health, Korea University Graduate School, Seoul, Korea, **3** Graduate School of Public Health, Yonsei University, Seoul, Korea, **4** College of Nursing and Health, Kongju National University, Gongju, Korea, **5** Division of Infectious Diseases, Department of Medicine, Samsung Medical Center, Sungkyunkwan University School of Medicine, Seoul, Korea, **6** Department of Internal Medicine, Severance Hospital, Yonsei University College of Medicine, Seoul, Korea, **7** Department of Infectious Diseases, Ajou University School of Medicine, Suwon, Korea

☯ These authors contributed equally to this work.
* jyheomd@ajou.ac.kr

## Abstract

Interferon (IFN) has been highlighted in several randomized controlled trials as an attractive therapeutic candidate based plausible mode of action, suppressed response in severe COVID-19, and inhibition of SARS-CoV-2 replication. This study investigated the efficacy and safety of IFN in patients with COVID-19 according to clinical severity. Randomized controlled trials evaluating the efficacy and safety of IFN (systemic or inhaled IFN-α, -β, and -λ) treatment in adult patients with COVID-19 were identified by systematically searching electronic databases until January 2023. Risk of bias were assessed using the Cochrane risk of bias tool, meta-analysis, and certainty of evidence grading were followed for the systematic review. We included 11 trials comprising 6,124 patients. Compared with exclusive standard care or placebo, IFN therapy did not provide significant clinical benefits for mortality at day 28 (pooled risk ratio [RR] = 0.86, 95% confidence interval [CI]: 0.62–1.18, 9 studies, low-certainty evidence) and progression to mechanical ventilation (pooled RR = 1.08, 95% CI: 0.81–1.43, 6 studies, low-certainty evidence) in patients with COVID-19. IFN therapy resulted in significantly increased hospital discharge on day 14 relative to the control arm (pooled RR = 1.29, 95% CI: 1.04–1.59). These results were inconsistent compared to other comparable outcomes such as recovery at day 14 and time to clinical improvement. The IFN-treated arm was as safe as the control arm, regardless of clinical severity (pooled RR = 0.87, 95% CI: 0.64–1.19, 9 studies, low-certainty evidence). In conclusion, IFN therapy was safe but did not demonstrate favorable outcomes for major clinical indices in patients with COVID-19, particularly those with higher than moderate severity. IFN therapy was not associated with worsening outcomes in patients with severe COVID-19. Future clinical trials should evaluate the clinical efficacy of IFN therapy in patients with mild COVID-19 or at an earlier stage.

**Data Availability Statement:** All relevant data are within the paper and its Supporting information files.

**Funding:** This research was supported by the National Evidence-based Collaborating Agency, Republic of Korea (grant number: NA22-008). There was no additional internal or external funding received for this study. The funding source had no role in the study design, data collection, and analysis, decision to publish, or manuscript preparation of the manuscript.

**Competing interests:** The authors have declared that no competing interests exist.

**Trial registration**: The protocol for this review was prospectively registered in the International Prospective Register of Systematic Reviews (PROSPERO) under the registration number CRD42022301413.

## Introduction

The coronavirus disease 2019 (COVID-19) pandemic is globally an ongoing public health crisis ever since the emergence of severe acute respiratory syndrome coronavirus 2 (SARS-CoV-2) in late 2019. Although vaccination is currently the most effective strategy to minimize the devastating impact of the COVID-19 pandemic, the continued emergence of new variant strains of SARS-CoV-2 such as Delta (B.1.617.2) and Omicron (B.1.1.529) have caused large outbreaks among highly vaccinated populations via immune escape [1, 2]. Several therapeutic options such as antiviral agents and monoclonal antibodies have been introduced to reduce the risk for progression to severe COVID-19. However, access to newly developed drugs in low- and middle-income countries is difficult due to high costs and limited supply [3–6]. Thus, alternative therapeutic options are required to improve the availability and affordability of COVID-19 medicines.

Interferons (IFN) are well-known therapeutics with antiviral activity and immunoregulatory properties. There are three types of IFN, classified by the type of receptor that mediates signaling. Of these, types I (IFN-α and IFN-β) and III (IFN-λ) IFN responses are markedly reduced in patients with severe COVID-19 [7, 8]. Although types I and III IFN share common properties, including induction by viral infection and signaling pathways, they signal via different receptors [9, 10]. Type I IFN receptors are ubiquitously expressed, whereas type III IFN receptors are preferentially expressed in epithelial cells in the lung, liver, and intestine. Thus, type III IFN is associated with lower inflammatory potency and fewer systemic side effects compared to type I IFN. These IFNs have been reported to exert favorable effects such as viral suppression in hepatitis B and C as well as lower mortality and faster improvement of chest radiograph in SARS and MERS [11–13]. Since the COVID-19 pandemic, IFNs have been evaluated in several randomized controlled trials as attractive therapeutic candidates due to their plausible mode of action, suppression of IFN activity in severe COVID-19, and inhibition of SARS-CoV-2 replication *in vivo* and *in vitro* [14]. Seminal studies demonstrated that systemic IFN-β was more likely to alleviate symptoms and shorten the duration of viral shedding in hospitalized patients with mild-to-moderate COVID-19 [15, 16]. However, a large randomized controlled trial (RCT) did not demonstrate that the combination therapy of IFN-β and remdesivir can lead to reduce the time to clinical recovery, compared to remdesivir alone in hospitalized patients with moderate-to-severe COVID-19 [17]. Additionally, the World Health Organization solidarity trial did not reveal favorable efficacy of IFN in hospitalized patients with moderate-to-severe COVID-19 [18]. Nonetheless, systemic IFN-α or -λ, inhaled IFN-β, and systemic IFN-β have still been evaluated in patients with COVID-19 of various severities [14]. Indeed, these agents will likely be tested in future clinical trials for newly emerging viral infections. However, the optimal timing and route of IFN administration and type of IFN used for the treatment of acute respiratory viral illnesses such as COVID-19 remain unclear.

Therefore, we performed a systematic review and meta-analysis to evaluate the efficacy and safety of types I and III IFN treatment in patients with COVID-19 with different clinical severities.

## Materials and methods

A systematic review of randomized controlled trials (RCTs) was conducted using a meta-analysis in accordance with the recommendations of the Cochrane Handbook and Preferred

Reporting Items for Systematic Review and Meta-analysis (PRISMA) statement 2020 [19]. The protocol for this review was prospectively registered in the International Prospective Register of Systematic Reviews (PROSPERO) under the registration number **CRD42022301413.**

## Search strategy

We systematically searched PubMed, Ovid-EMBASE, CENTRAL, and Korean databases (KMBASE) until June 11, 2021. Ongoing trials or pre-published articles were excluded. For completeness, reference lists of relevant primary and review articles were searched manually. Since new evidence on treatments for COVID-19 is continuously produced, the search was updated on the 10th day of each month, starting from August 2021 to March 2022, and January 2023. We systematically searched Ovid-MEDLINE for updates. The complete electronic search strategy for each database is presented in S1 File.

## Eligibility criteria and study selection

Articles that met the following requirements were considered: 1) patients were adults with COVID-19; 2) interventions used IFN (IFN-α, β, and λ) 3); the comparator was placebo or standard-of-care (SOC) treatment; 4) outcomes included 28-day mortality, progression to mechanical ventilation, serious adverse events, recovery and hospital discharge on day 14, time to clinical improvement, and length of hospital stay; and 5) the study was designed as a RCT. Only English and Korean studies were included in this meta-analysis. Two authors (SR and SY) independently and in duplicate evaluated publications for inclusion based on the title and abstract and then reviewed relevant full-text articles. Disagreements during the review process were addressed by consensus, with the involvement of a third author (JH).

## Risk of bias assessment and data extraction

Two authors (SR and DK) independently assessed the quality of the selected studies using the Cochrane risk of bias tool [20]. Disagreements were resolved by consensus, with the involvement of a third author (SY). Two review authors (DK and SR) extracted the information from each included trial. These evaluations were performed independently and yielded separate results. Disagreements were resolved by discussion and a third opinion (MC). The following information was included in the data extraction form: first author, publication date, study design, characteristics of study participants, IFN therapeutic type, and outcomes.

To align the included research as a single figurative criterion, data were collected from the electronic supplementary material or, when possible, using the intention-to-treat (ITT) principle (if not defined in the original article). To obtain additional information, we contacted the corresponding authors of included trials that had insufficient information.

## Rating certainty of evidence

Certainty of evidence was graded using the Grading of Recommendations, Assessment, Development, and Evaluation (GRADE) approach for primary outcomes and serious adverse events [21]. The primary outcomes included mortality on day 28, progression to mechanical ventilation, and serious adverse events. Recovery/hospital discharge on day 14, time to clinical improvement, and length of hospital stay were classified as secondary outcomes.

## Data synthesis and statistical analysis

For each included trial, continuous outcomes were presented as mean differences or hazard ratios (HRs) with inverse-variance random-effects analysis and dichotomous outcomes as risk

ratios (RRs) with Mantel–Haenszel random-effects analysis and 95% confidence intervals (CIs) for all outcome measures. Heterogeneity among trials was explored by inspecting forest plots and calculating $I^2$ statistics.

We conducted a pre-planned subgroup analysis according to clinical severity. Based on the National Institute of Allergy and Infectious Diseases Ordinal Scale of COVID-19 Severity, clinical severity was defined as mild (not hospitalized or hospitalized without requiring supplemental oxygen and ongoing medical care), moderate (hospitalized requiring ongoing medical care without supplemental oxygen), or severe (hospitalized requiring any types of supplemental oxygen, including low- or high-flow oxygen devices, noninvasive ventilation, and invasive mechanical ventilation [22]. If results of outcome parameters were not clearly reported according to clinical severity, data were classified as a separate group (moderate-to-severe group). As for serious adverse events, we defined it as generally accepted definition: 1) death or life-threatening event, 2) hospitalization (initial or prolonged), 3) Disability or permanent damage, 4) Congenital anomaly or birth defect, 5) Required intervention to prevent permanent impairment, 6) Other serious medical events such as anaphylaxis, seizure or emergency room visit. Statistical analyses were performed using Review Manager software version 5.4. Stratification details are available in S2 File. Publication bias for 28-day mortality was assessed through visual inspection of funnel plot. For data with an asymmetric funnel plot, Egger's linear regression test was additionally performed using Stata version 14.

## Results

### Description of included studies

A total of 8,305 articles were retrieved from the databases on 11 June 2021. After excluding duplicates, 7,170 articles were identified. After the living search update, total of 7,826 records were screened as of 23 January 2023. Based on the selection criteria, 137 articles were selected for full-text review. A final total of 11 RCTs comprising 6,124 patients were included in this systematic review [16–18, 23–30]. Details of the study selection and review flowchart are presented in Fig 1. Of the 11 selected studies, clinical severity of COVID-19 in enrolled participants was reclassified based on pre-defined severity, as follows: two RCTs for mild cases (Feld 2021, Jagannathan 2021) [27, 28], two RCTs for moderate and severe cases (Kalil 2021, Pan 2021) [17, 18], one RCT for moderate-to-severe cases (Monk 2021) [16], five RCTs for only severe cases (Bhushan 2021, Darazam 2021, Pandit 2021, Davoudi-Monfared 2020, Rahmani 2020) [24–26, 29, 30], and one RCT for moderate-to-severe and severe cases (Ader 2021) [23]. Two studies [16, 23] (of Monk 2021 and Ader 2021) only reported pooled analyses of outcome parameters in the moderate-to-severe group. The characteristics of the included studies are presented in Table 1. The results of the risk of bias summary are presented in S3 File. Most studies had a low risk of bias. The GRADE evidence profiles, and summary of the findings are presented in Table 2.

### Primary outcomes: 28-day mortality

Nine studies, comprising 2,895 cases in the IFN arm and 2,869 controls in the placebo or SOC arm, investigated the effect of IFN treatment on 28-day mortality. Compared with the control arm, IFN therapy tended to decrease mortality at day 28, but this was not statistically significant (pooled RR = 0.86, 95% confidence interval [CI]: 0.62–1.18, $I^2$ = 40%, low certainty evidence) (Fig 2). Subgroup analysis according to clinical severity of COVID-19 revealed that IFN treatment failed to reduce 28-day mortality in any sub-group (mild group, not estimable; moderate group, pooled RR = 0.92, 95% CI: 0.35–2.43, $I^2$ = 7%; moderate-to-severe group, pooled RR = 0.60, 95% CI: 0.11–3.45, $I^2$ = 31%; severe group, pooled RR = 0.84, 95% CI: 0.57–

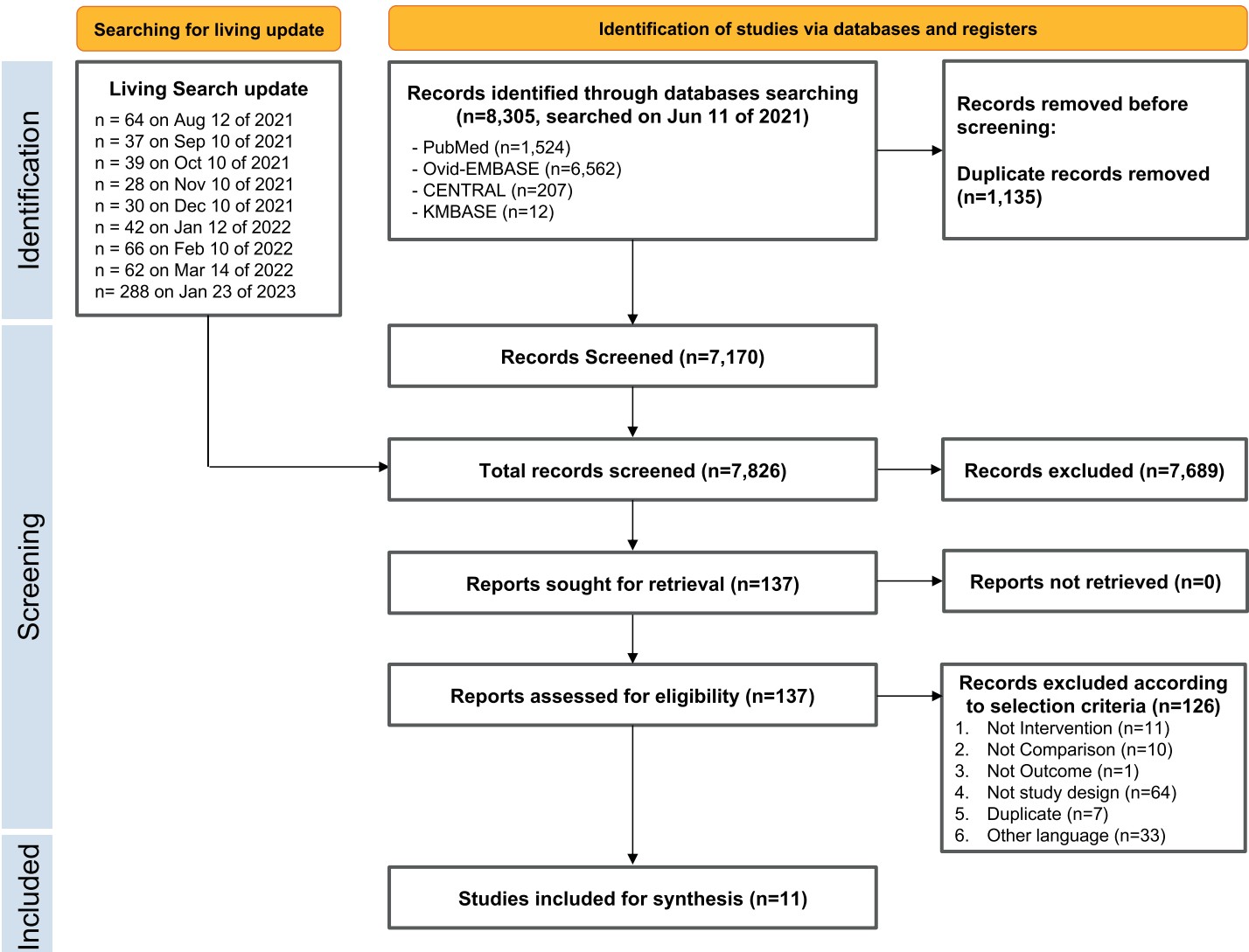

**Fig 1. Preferred reporting items for systematic reviews and meta-analyses (PRISMA) study flowchart.**

1.25, $I^2$ = 59%). Although IFN therapy tended to reduce 28-day mortality in the subgroup of patients with moderate-to-severe COVID-19, the difference did not reach statistical significance (pooled RR = 0.60, 95% CI: 0.11–3.45, $I^2$ = 31%). The publication bias of the included studies was evaluated to be low risk in the domain of 28-day mortality. Although the funnel plot was asymmetric, Egger's test did not reveal statistically significant publication bias ($p$ = 0.535) in S4 File.

## Primary outcomes: Progression to mechanical ventilation

Progression to mechanical ventilation as an outcome parameter was reported in six studies comprising 2,744 IFN-treated cases and 2,739 controls. No statistically significant differences were observed in the rate of progression to mechanical ventilation between the IFN-treated and control arms (pooled RR = 1.08, 95% CI: 0.81–1.43, $I^2$ = 38%, low certainty evidence) (Fig 3). Further subgroup analysis based on clinical severity revealed no significant differences

**Table 1. Baseline study characteristics of published randomized controlled trials of interferon.**

| Author (Year) /Study site [Ref]· | Study design | Population (n = patients included in analysis) and comorbidities (n) | Intervention arm (No.) | Control arm (No.) | INF dose and schedule | Median age, (years) | COVID-19 severity | Outcomes |
|---|---|---|---|---|---|---|---|---|
| Feld (2021) / Canada [27] | Double-blind RCT | Outpatients (59) Hypertension (6) Diabetes (3) Heart disease (2) | Peg-IFN λ-1a | placebo | SC, 180 μg, single dose | I: 48.0 C: 39.0 | Mild | • Proportion of viral negativity at day 7 by quantitative SARS-CoV-2 PCR • incidence of treatment-emergent SAE by day 14 |
| Jagannathan (2021) /U.S. [28] | Single-blind RCT | Outpatients (120) Hypertension (14) Diabetes (12) Asthma (4) Heart disease (4) | Peg-IFN λ-1a | placebo | SC, 180 μg, single dose | I: 37.0 C: 34.0 | Mild | •Time to two consecutive negativity for SARS-CoV-2 PCR • Time to clinical improvement |
| Pan* (2021) /International Solidarity trial [18] | Open-label RCT | Hospitalized patients (972) | IFN β-1a ± SOC ± LPV/r | SOC ± LPV/r | SC, 44 μg every other day, three doses; IV 10 μg daily for 6 days | N/A | Moderate | • In-hospital mortality • the initiation of mechanical ventilation • hospitalization duration |
| Kalil[†] (2021) /International ACTT-3 [17] | Double-blind RCT | Hospitalized patients (152) | IFN β-1a + SOC + RDV | SOC (±steroid) +RDV | SC, 44 μg every other day, four doses | I: 58.3 C: 59.1 | Moderate | • Time to recovery by day 28 days • Odds of clinical improvement • Time to clinical improvement • Incidence and duration of new supplemental oxygen use, non-invasive ventilation or high-flow oxygen, and invasive ventilation • duration of hospitalization up to day 29 |
| Ader[‡] (2021) /France, DisCoVeRy [23] | Open-label RCT | Hospitalized patients (182) | IFN β-1a + SOC + LPV/r | SOC + LPV/r | SC, 44 μg on days 1, 3, 6 | I: 64.0 C: 63.0 | Moderate-to-Severe | • Clinical status at day 15 • Time to clinical improvement • Hospital discharge until day 29 • Time to hospital discharge until day 29 • 29-day mortality |
| Monk (2021) /UK.[16] | Double-blind RCT | Hospitalized patients (98) Hypertension (29) Chronic lung disease (23) Cardiovascular disease (13) Diabetes (12) Cancer (1) | IFN β-1a | placebo | Inhalation, 6 MIU once daily for up to 14 days | I: 57.8 C: 56.5 | Moderate-to-Severe | • Change of clinical status |
| Darazam (2021) /Iran, COVIFERON [24] | Open-label RCT | Hospitalized patients (60) Diabetes (14) Hypertension (20) Chronic heart disease (10) Chronic kidney disease (5) Cancer (1) | (1) IFN β-1a + SOC (2) IFN β-1b + SOC | SOC | (1) IFN β-1a, SC, 44 μg on days 1, 3, 6 (2) IFN β-1b 8 MIU on days 1, 3, 6 | I (1): 71.5 I (2): 65.0 C: 76.0 | Severe | • Time to clinical improvement • Mortality until day 21 • Hospital discharge • Death |

(*Continued*)

**Table 1.** (Continued)

| Author (Year) /Study site [Ref]· | Study design | Population (n = patients included in analysis) and comorbidities (n) | Intervention arm (No.) | Control arm (No.) | INF dose and schedule | Median age, (years) | COVID-19 severity | Outcomes |
|---|---|---|---|---|---|---|---|---|
| Ader[‡] (2021) /France, DisCoVeRy [23] | Open-label RCT | Hospitalized patients (108) | IFN β-1a + SOC + LPV/r | SOC + LPV/r | SC, 44 µg on days 1, 3, 6 | I: 64.0 C: 63.0 | Severe | • Time to clinical improvement<br>• Hospital discharge until day 29<br>• Time to hospital discharge until day 29<br>• 29-day mortality |
| Davoudi-Monfared (2020)/Iran [26] | Open-label RCT | Hospitalized patients (81) Hypertension (31) Diabetes (22) Heart disease (23) Endocrine disorder (12) Cancer (9) | IFN β-1a + SOC | SOC (LPV/r or ATV/r + HCQ for 7–10 days) | SC, 44 µg every other day for two weeks | I: 56.0 C: 59.5 | Severe | • Time to clinical improvement<br>• Duration of mechanical ventilation<br>• Duration of hospital stay<br>• Length of ICU stay<br>• 28-day mortality |
| Pan[*] (2021) /International Solidarity trial [18] | Open-label RCT | Hospitalized patients (3128) | IFN β-1a ± SOC ± LPV/r | SOC ± LPV/r | SC, 44 µg every other day, three doses; IV 10 µg daily for 6 days | N/A | Severe | • In-hospital mortality<br>• the initiation of mechanical ventilation<br>• hospitalization duration |
| Kalil[†] (2021) /International ACTT-3 [17] | Double-blind RCT | Hospitalized patients (817) | IFN β-1a + SOC + RDV | SOC (±steroid) +RDV | SC, 44 µg every other day, four doses | I: 58.3 C: 59.1 | Severe | • Time to recovery by day 28 days<br>• Odds of clinical improvement<br>• Time to clinical improvement<br>• Incidence and duration of new supplemental oxygen use, non-invasive ventilation or high-flow oxygen, and invasive ventilation<br>• duration of hospitalization up to day 29 |
| Rahmani (2020) /Iran [30] | Open-label RCT | Hospitalized patients (66) Hypertension (37) Diabetes (21) Heart disease (22) Asthma 3 COPD (3) Cancer (2) | IFN β-1b | SOC (LPV/r or ATV/r + HCQ for 7–10 days) | SC, 250 µg every other day for two weeks | I: 60 C: 61 | Severe | • Time to clinical improvement<br>• Side effects related to IFN therapy and other adverse events during the study period |
| Bhushan (2021) /India [25] | Open-label RCT | Hospitalized patients (242) Not available for comorbidity | Peg-IFN α-2b + SOC | SOC (± HCQ ± RDV ± steroid) | SC, 1 µg/kg, single dose | I: 49.6 C: 50.1 | Severe | • Clinical improvement at day 11<br>• clinical status at Days 8, 11 and 15<br>• Proportion of subjects with AEs<br>• Qualitative PCR for SARS-CoV-2 |

(*Continued*)

**Table 1.** (Continued)

| Author (Year) /Study site [Ref]· | Study design | Population (n = patients included in analysis) and comorbidities (n) | Intervention arm (No.) | Control arm (No.) | INF dose and schedule | Median age, (years) | COVID-19 severity | Outcomes |
|---|---|---|---|---|---|---|---|---|
| Pandit (2021) /India [29] | Open-label RCT | Hospitalized patients (39) Not available for comorbidity | Peg-IFN α-2b + SOC | SOC (± HCQ ± steroid) | SC, 1 µg/kg, single dose | N/A | Severe | • Clinical improvement at day 15 • Proportion of subjects with AEs • Occurrence and duration of supplemental O2/MV • Duration of hospitalization |

*Pan et al. study included the patients with moderate (n = 972) and severe (n = 3128) COVID-19. This study did not provide the information for comorbidities based on different severity group. The study patients had comorbidities of diabetes (n = 1026), heart disease (n = 883), chronic lung disease (n = 223), asthma (n = 172) and chronic liver disease (n = 33).

†Kalil et al. study included the patients with moderate (n = 152) and severe (n = 817) COVID-19. This study did not provide the information for comorbidities based on different severity group. The study patients had comorbidities of hypertension (n = 559), obesity (n = 555), diabetes (n = 352), depression/psychotic disorder (n = 170), coronary artery disease (n = 126), asthma (n = 122), chronic kidney disease (n = 112) and chronic respiratory disease (n = 105).

‡Ader et al. study included the patients with moderate to severe (n = 182) and severe (n = 108) COVID-19. This study did not provide the information for comorbidities based on different severity group. The study patients had comorbidities of chronic cardiac disease (n = 75), chronic pulmonary disease (n = 50), chronic kidney disease (n = 12), cancer (n = 16), obesity (n = 87) and diabetes (n = 62).

Abbreviations: C, control; COPD, chronic obstructive pulmonary disease; I, intervention; IFN, interferon; HCQ, hydroxychloroquine; LPV/r, lopinavir/ritonavir; RCT, randomized controlled trial; RDV, remdesivir; SpO₂, peripheral oxygen saturation; PRISMA, Preferred Reporting Items for Systematic Reviews and Meta-Analysis; SC, subcutaneous; SOC, standard of care including antibiotics, antiviral agents, corticosteroids, vasopressor support, and anticoagulants

between the intervention and control arms in any subgroup (moderate-to-severe group: pooled RR = 1.00, 95% CI: 0.84–1.18, $I^2$ = 0%; severe group: pooled RR = 1.23, 95% CI: 0.70–2.16, $I^2$ = 59%).

## Primary outcomes: Serious adverse events

Serious adverse events (SAEs) were reported in nine studies comprising 972 IFN-treated cases and 954 controls. Large differences were noted in the incidence of SAE among the nine studies, ranging from 3.3% in Feld et al. (2021) [27] to 65% in Alavi Darazam et al. (2021) [24]. Studies including cases with a higher severity were associated with more SAEs. A study by Kalil et al. (2021) [17] demonstrated that IFN therapy was associated with a significantly increased incidence of SAEs. Compared to the control arm, IFN therapy did not lead to a significant increase in the development of SAEs (pooled RR = 0.87, 95% CI: 0.64–1.19, $I^2$ = 66%, low certainty evidence) (Fig 4). Subgroup analysis based on clinical severity revealed no statistically significant differences in the incidence of SAEs between IFN-treated cases and controls for all groups (mild group: pooled RR = 1.00, 95% CI: 0.21–4.82, $I^2$ = 0%; moderate-to-severe group: pooled RR = 1.00, 95% CI: 0.79–1.27, $I^2$ = 29%; severe group: pooled RR = 0.82, 95% CI: 0.42–1.63, $I^2$ = 83%).

## Secondary outcomes

Secondary outcomes included recovery and hospital discharge on day 14, time to clinical improvement, and length of hospital stay. Recovery on day 14 was evaluated as an outcome parameter in four studies with 674 IFN-treated cases and 674 controls. IFN therapy did not result in a clinical benefit for recovery on day 14 (pooled RR = 1.03, 95% CI: 0.94–1.12, $I^2$ = 63%) (S1 Fig in S1 File). A study by Monk et al. (2021) revealed that IFN therapy administered

**Table 2. GRADE summary of findings table of mortality, progression to invasive mechanical ventilation, and serious adverse events.**

| Outcomes | Sub-groups | Anticipated absolute effects* (95% CI) | | Relative effect (95% CI) | of participants (studies) | Certainty of the evidence (GRADE) |
|---|---|---|---|---|---|---|
| | | Risk with standard care/ placebo | Risk with Interferon | | | |
| Mortality at 28 days | Total | 98 per 1,000 | 84 per 1,000 (61 to 116) | RR 0.86 (0.62 to 1.18) | 5764 (9 RCTs) | ⊕⊕◯◯ Low[a,b] |
| | Mild | 0 per 1,000 | 0 per 1,000 (0 to 0) | not estimable | 60 (1 RCT) | ⊕⊕◯◯ Low[c] |
| | Moderate | 23 per 1,000 | 21 per 1,000 (8 to 57) | RR 0.92 (0.35 to 2.43) | 1124 (2 RCTs) | ⊕⊕◯◯ Low[b,d] |
| | Moderate-to-severe | 49 per 1,000 | 29 per 1,000 (5 to 168) | RR 0.60 (0.11 to 3.45) | 283 (2 RCTs) | ⊕⊕◯◯ Low[b,d] |
| | Severe | 122 per 1,000 | 103 per 1,000 (70 to 153) | RR 0.84 (0.57 to 1.25) | 4297 (7 RCTs) | ⊕⊕◯◯ Low[a,b] |
| Progression to mechanical ventilation | Total | 102 per 1,000 | 110 per 1,000 (83 to 146) | RR 1.08 (0.81 to 1.43) | 5483 (6 RCTs) | ⊕⊕◯◯ Low[a,b] |
| | Moderate-to-severe | 95 per 1,000 | 95 per 1,000 (80 to 112) | RR 1.00 (0.84 to 1.18) | 4926 (3 RCTs) | ⊕⊕⊕⊕ High |
| | Severe | 165 per 1,000 | 203 per 1,000 (115 to 356) | RR 1.23 (0.70 to 2.16) | 557 (5 RCTs) | ⊕⊕◯◯ Low[a,b] |
| Serious adverse events | Total | 223 per 1,000 | 194 per 1,000 (143 to 266) | RR 0.87 (0.64 to 1.19) | 1926 (9 RCTs) | ⊕⊕◯◯ Low[a,b] |
| | Mild | 33 per 1,000 | 33 per 1,000 (7 to 161) | RR 1.00 (0.21 to 4.82) | 180 (2 RCTs) | ⊕⊕◯◯ Low[b,d] |
| | Moderate-to-severe | 235 per 1,000 | 235 per 1,000 (186 to 299) | RR 1.00 (0.79 to 1.27) | 1263 (3 RCTs) | ⊕⊕⊕◯ Moderate[b] |
| | Severe | 264 per 1,000 | 216 per 1,000 (111 to 430) | RR 0.82 (0.42 to 1.63) | 483 (5 RCTs) | ⊕◯◯◯ Very low[a,b,e] |

GRADE Working Group grades of evidence

**High certainty**: We are confident that the true effect lies close to that of the estimate of the effect.

**Moderate certainty**: We are moderately confident in the effect estimate; the true effect is likely to be close to the estimate of the effect, but there is a possibility that it is substantially different.

**Low certainty**: Our confidence in the effect estimate is limited; the true effect may be substantially different from the estimate of the effect.

**Very low certainty**: We have very little confidence in the effect estimate; the true effect is likely to be substantially different from the estimate of the effect.

The risk in the intervention group (and its 95% CI) was based on the assumed risk in the comparison group and the relative effect of the intervention (and its 95% CI).

[a] Risk of bias downgraded by 1 level

[b] Imprecision downgraded by 1 level due to a wide confidence interval consistent with the possibility for benefit and the possibility for harm

[c] Imprecision downgraded by 2 level due to only one study with low number of events

[d] Imprecision downgraded by 1 level due to wide confidence interval

[e] Inconsistency downgraded by 1 level due to the confidence intervals between studies do not overlap

Abbreviation: CI, Confidence interval; RR, Risk ratio

via inhalation resulted in a significantly higher recovery rate in patients with moderate-to-severe COVID-19 (RR = 1.72, 95% CI: 1.09–2.70) [16]. Three studies, which included 123 IFN-treated cases and 122 controls, reported hospital discharge rates on day 14. The pooled analysis revealed that IFN therapy significantly increased the rate of hospital discharge on day 14 relative to the control arm (pooled RR = 1.29, 95% CI: 1.04–1.59, $I^2$ = 17%) (S2 Fig in S1

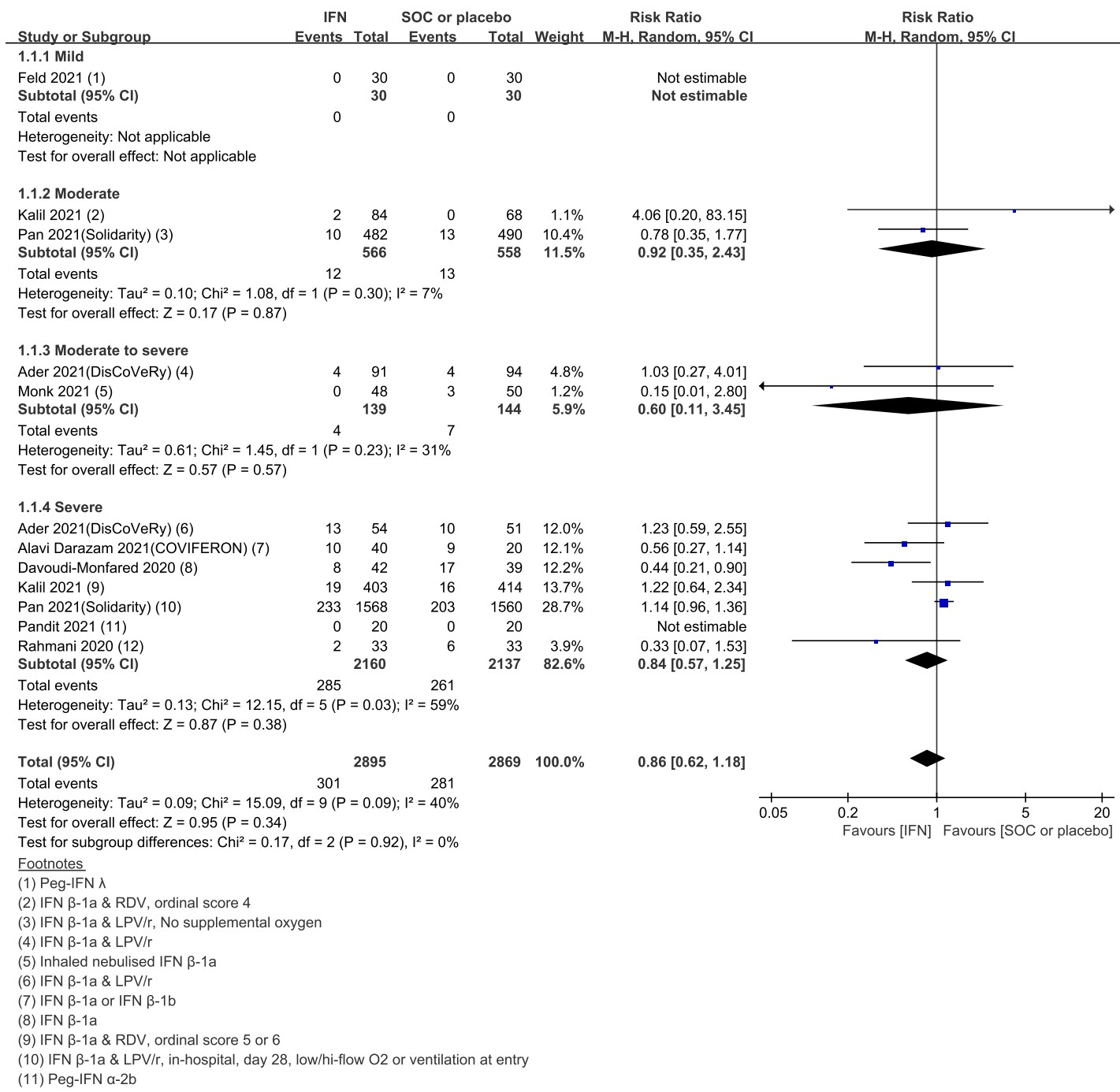

**Fig 2. Forest plot of 28-day mortality.** Forest plot presenting the risk ratio (RR) for mortality between the interferon (IFN)-treated and control arms. Meta-analysis of nine randomized controlled trials (RCTs) comprising 5,764 patients revealed that IFN therapy failed to reduce 28-day mortality compared to the control arm, regardless of clinical severity (overall group, pooled RR = 0.86, 95% CI: 0.62–1.18, $I^2$ = 40%; mild group, not estimable; moderate group, pooled RR = 0.92, 95% CI: 0.35–2.43; $I^2$ = 7%; moderate-to-severe group, pooled RR = 0.60, 95% CI: 0.11–3.45, $I^2$ = 31%; severe group, pooled RR = 0.84, 95% CI: 0.57–1.25, $I^2$ = 59%).

File). Subgroup analysis by clinical severity revealed that this significant result was consistently observed only in the severe subgroup (pooled RR = 1.48, 95% CI: 1.13–1.94, $I^2$ = 0%) but not in the moderate-to-severe subgroup (RR = 1.10, 95% CI: 0.85–1.44, $I^2$ = not applicable). Time to clinical response as an outcome parameter was analyzed in seven studies comprising 880

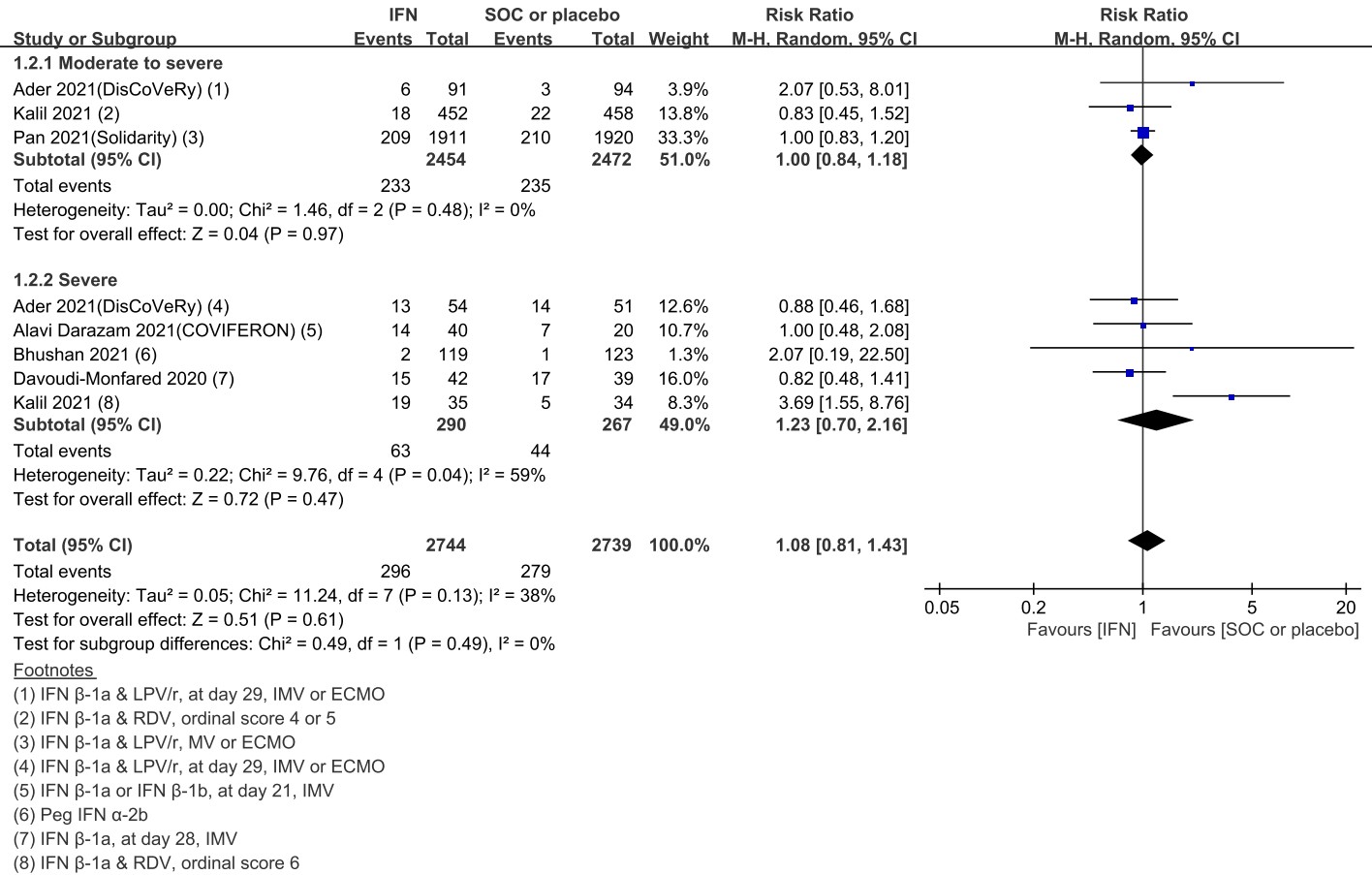

**Fig 3. Forest plot of progression to mechanical ventilation.** Forest plot presenting the risk ratio (RR) for progression to mechanical ventilation between the interferon (IFN)-treated and control arms. Meta-analysis of six randomized controlled trials (RCTs) comprising 5,483 patients revealed no significant difference in progression to mechanical ventilation between the IFN-treated and control arms (overall group: pooled RR = 1.08, 95% CI: 0.81–1.43, $I^2$ = 38%; moderate-to-severe group: RR = 1.00, 95% CI: 0.84–1.18, $I^2$ = 0%; severe group: RR = 1.23, 95% CI: 0.70–2.16, $I^2$ = 59%).

IFN-treated cases and 859 controls. Although IFN therapy tended to decrease the time to clinical response, this was not statistically significant (pooled RR = -0.73, 95% CI: -1.48–0.02, $I^2$ = 48%) (S3 Fig in S1 File). Subgroup analysis revealed no significant differences between IFN-treated and control arms (mild subgroup: RR = -1.0, 95% CI: -4.91–2.91; moderate subgroup: RR = 0.00, 95% CI: -1.12–1.12; moderate-to-severe subgroup: RR = -1.00, 95% CI: -3.36–1.36; severe subgroup: pooled RR = -0.95, 95% CI: -1.48–0.02). IFN therapy had no favorable effect on length of hospital stay, regardless of clinical severity (pooled RR = 0.40, 95% CI: -0.74–1.54, $I^2$ = 87%) (S4 Fig in S1 File).

## Discussion

This meta-analysis and systematic review analyzed 11 RCTs to evaluate the efficacy and safety of IFN therapy for the treatment of COVID-19 compared with SOC or placebo. Overall, IFN therapy did not have significant clinical benefits for mortality or progression to mechanical ventilation. IFN therapy demonstrated a tendency to reduce mortality, but this did not reach statistical significance. Further, IFN therapy did not exert a beneficial influence on preventing disease progression to mechanical ventilation. These results remained unchanged after subgroup analysis according to clinical severity. In addition, IFN therapy did not improve

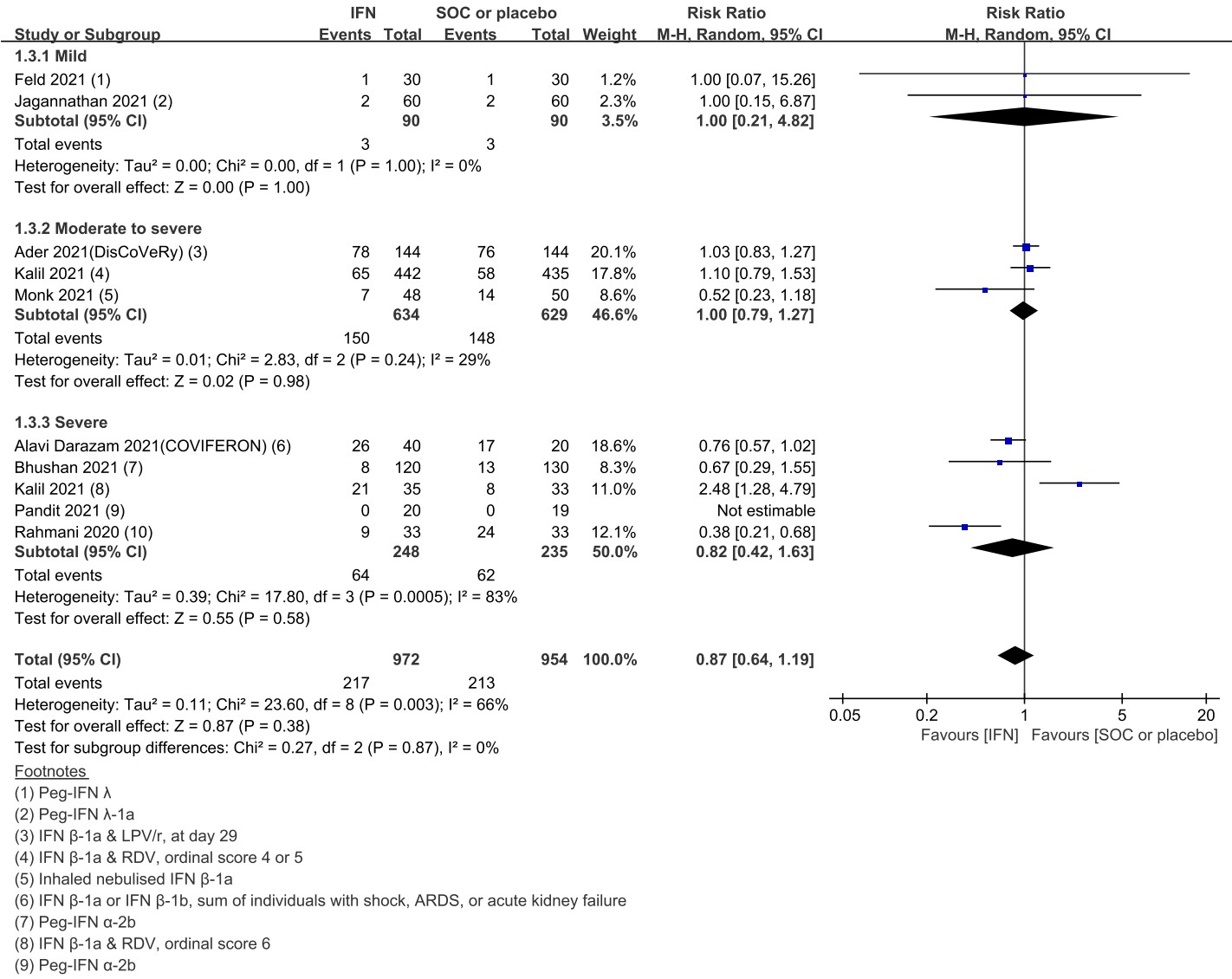

**Fig 4. Forest plot of serious adverse events.** Forest plot presenting the risk ratio (RR) of serious adverse events between the interferon (IFN)-treated and control arms. Meta-analysis of nine randomized controlled trials (RCTs) comprising 1,926 patients revealed that IFN therapy did not lead to a significant increase in the development of serious adverse events compared to the control arm (pooled RR = 0.87, 95% CI: 0.64–1.19; $I^2$ = 66%).

recovery on day 14 and did not shorten the time to clinical improvement or duration of hospital stay in any subgroup.

We evaluated 11 RCTs including two RCTs of patients with mild-severity COVID-19. However, these results were primarily obtained from RCTs of patients with moderate-to-severe or severe disease. Given that disease progression to severe illness in the natural course of COVID-19 typically occurs within 7–14 days after symptom onset, a large number of clinical trials related to IFN were performed at the phase of inflammatory response in the host after a decline in SARS-CoV-2 viral load. Two early studies provided a theoretical background to the role of IFN in defense mechanisms against SARS-CoV-2 infection. First, some patients with life-threatening COVID-19 pneumonia harbored genetic defects that were involved in

the production of type I IFN, suggestive of an inborn error of the type I IFN response [31]. Second, approximately 10% of patients with critical COVID-19 had neutralizing autoantibodies to type I IFN, which may have impaired receptor binding of type I IFN and activation of downstream pathways [32]. As a result, recombinant IFN has been actively tested in numerous clinical trials as a promising therapeutic for COVID-19 [14]. However, our findings suggest that IFN therapy does not exert clinical benefits in patients with COVID-19 of higher than moderate severity.

In contrast, IFN therapy was not associated with worsening outcomes in patients with severe COVID-19. Several preclinical studies reported that the type I IFN response co-occurs with proinflammatory cytokine responses such as TNF-α and IL-1β, which is a major feature of severe COVID-19 [33–35]. Upregulation of type I IFN at a later stage of COVID-19 may play a major role in the deterioration of inflammatory responses in the progression to severe COVID-19. This finding suggests that a delayed IFN response causes lung damage and pneumonia. Therefore, the time of administration should be carefully considered for the therapeutic use of IFN. However, the results of our meta-analysis revealed that IFN administration in the late phase or severe stage of COVID-19 was not associated with poor clinical outcomes.

Although most outcome parameters did not exhibit show a clinically beneficial effect of IFN therapy in patients with COVID-19, patients receiving IFN were more likely to be discharged from hospital on day 14. This inconsistent result was primarily derived from studies by Davoudi-Monfared et al. (2020) and Rahmani et al. (2020) [26, 30] However, hospital discharge on day 14 was comparable to recovery on day 14 and time to clinical improvement. Given that these comparative targets did not provide a significantly favorable outcome, they cannot be considered clinically significant.

We assessed the safety of IFN, defined as SAEs. IFN administration is usually accompanied by a wide range of adverse events from flu-like symptoms to autoimmune diseases such as psoriasis or psychiatric symptoms, such as aggressive behavior. However, many adverse events following IFN therapy are associated with dose-dependent responses [36]. Although IFN therapy in patients with severe COVID-19 seems to be associated with more SAEs, our analysis revealed that IFN treatment was as safe as the control arm, regardless of clinical severity.

Previous reviews have several limitations with regard to the evaluation of IFN efficacy in the treatment of COVID-19. Previous meta-analyses were restricted to IFN-β [37, 38], and data synthesis of the included studies did not consider the study design, such as retrospective observational/cohort and RCT, leading to imprecision in estimating the effect estimate of IFN [39]. A major limitation of previous reviews was that they evaluated the efficacy of IFN therapy in heterogeneous patients with different severities of COVID-19 [37, 39, 40]. Given that the benefits and harms of IFN therapy may theoretically differ in the earlier or later stages of SARS-CoV-2 infection, it would be more reasonable to present effect size of IFN therapy according to the clinical stage of COVID-19. Thus, our meta-analysis presents overall efficacy of IFN therapy in patients with COVID-19 in addition to severity-specific efficacy. This may facilitate identification of the optimal timing of IFN administration in future clinical trials.

This study has several limitations. First, we were unable to sufficiently evaluate the clinical benefits of IFN therapy in patients with mild COVID-19. Although two RCTs on mild COVID-19 were included, the primary outcome in these studies was microbiologic outcome that assessed time to viral negativity or proportion of viral negativity at day 7 based on SARS-CoV-2 PCR [27, 28]. Thus, we also evaluated the clinical efficacy of IFN therapy for hospitalization or emergency room visits in outpatients with mild COVID-19. However, IFN therapy did not lead to significantly less hospitalization or emergency room visits in patients with mild COVID-19 (pooled RR = 0.85, 95% CI: 0.26–2.83, $I^2$ = 40%) (S5 Fig in S1 File). The sample sizes in studies by Feld et al. [27] (n = 60) and Jagannathan et al. [28] (n = 120) were too small,

thus precluding evaluation of the clinical benefits of IFN therapy for patients with mild COVID-19. Feld et al showed that IFN therapy could significantly reduce duration of viral shedding, whereas the study of Jagannathan did not demonstrate it. It means that lowering viral shedding can have potential to prevent clinical worsening. Thus, future clinical trial needs to demonstrate whether IFN therapy can reduce viral load in patients with mild COVID-19. Second, most of the findings, with the exception of outcomes of hospital discharge on day 14, resulted from moderate-to-high heterogeneity ($I^2 > 30\%$). This heterogeneity may be explained by the different doses and schedules of IFN, type of IFN, age group, comparators, and clinical severity. In this regard, subgroup analyses based on clinical severity were performed to address the expected heterogeneity.

In conclusion, IFN therapy was safe but did not demonstrate significant clinical benefits for mortality, progression to mechanical ventilation, and recovery in patients with COVID-19 with higher than moderate severity. In contrast, IFN therapy was not associated with worsening outcomes in patients with severe COVID-19. Future clinical trials should evaluate the clinical efficacy of IFN therapy in patients with mild COVID-19 or at an earlier stage.

## Supporting information

**S1 Table. Patient severity stratification.**
(PDF)

**S1 File. Supplementary figures.**
(PDF)

**S2 File. Search strategy.**
(PDF)

**S3 File. Risk of bias.**
(PDF)

**S4 File. Publication bias.**
(TIF)

**S5 File. PRISMA 2020 checklist.**
(PDF)

## Acknowledgments

The authors would like to thank the Task Force Members (Committee on the Establishment of Clinical Guidelines) for Emerging Infectious Diseases of the Korean Society of Infectious Diseases (KSID) for their help in developing this paper.

## Author Contributions

**Conceptualization:** Dae-Hyup Koh, Jung Yeon Heo.

**Data curation:** Seungeun Ryoo, Dae-Hyup Koh, Su-Yeon Yu, Miyoung Choi, Kyungmin Huh, Joon-Sup Yeom, Jung Yeon Heo.

**Formal analysis:** Seungeun Ryoo, Dae-Hyup Koh, Su-Yeon Yu, Miyoung Choi, Jung Yeon Heo.

**Methodology:** Seungeun Ryoo, Dae-Hyup Koh.

**Writing – original draft:** Seungeun Ryoo, Dae-Hyup Koh, Jung Yeon Heo.

**Writing – review & editing:** Seungeun Ryoo, Dae-Hyup Koh, Su-Yeon Yu, Miyoung Choi, Kyungmin Huh, Joon-Sup Yeom, Jung Yeon Heo.

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
