## [Decision Letter · Decision Letter 0]

15 Feb 2023

PONE-D-22-21005Clinical efficacy and safety of interferon (Type I and Type III) therapy in patients with COVID-19: A systematic review and meta-analysis of randomized controlled trialsPLOS ONE

Dear Dr. Heo,

Thank you for submitting your manuscript to PLOS ONE. After careful consideration, we feel that it has merit but does not fully meet PLOS ONE’s publication criteria as it currently stands. Therefore, we invite you to submit a revised version of the manuscript that addresses the points raised during the review process.

We look forward to receiving your revised manuscript.

Kind regards,

Ricardo Ney Oliveira Cobucci, Ph.D

Academic Editor

PLOS ONE

Journal Requirements:

“This research was supported by the National Evidence-based Collaborating Agency, Republic of Korea (grant number: NA22-008). The funding source had no role in the study design, data collection, and analysis, decision to publish, or manuscript preparation of the manuscript.”

Please provide an amended statement that declares *all* the funding or sources of support (whether external or internal to your organization) received during this study, as detailed online in our guide for authors at http://journals.plos.org/plosone/s/submit-now.  Please also include the statement “There was no additional external funding received for this study.” in your updated Funding Statement. Please include your amended Funding Statement within your cover letter. We will change the online submission form on your behalf.

‘This research was supported by the National Evidence-based Collaborating Agency, Republic of Korea (grant number: NA22-008). The funding source had no role in the study design, data collection, and analysis, decision to publish, or manuscript preparation of the manuscript.”

“This research was supported by the National Evidence-based Collaborating Agency, Republic of Korea (grant number: NA22-008). The funding source had no role in the study design, data collection, and analysis, decision to publish, or manuscript preparation of the manuscript.”

Additional Editor Comments (if provided):

Dear authors,

as you can see, the reviewers have requested minor revisions to your manuscript. We are certainly willing to reconsider a revised submission, but please know that this is not preliminary acceptance of your paper. When returning your revised manuscript, please be sure to include a point-by-point summary of the suggestions of the reviewers that specifies how and where in the text you have addressed the suggestions.

Reviewers' comments:

Reviewer's Responses to Questions

**Comments to the Author**

1. Is the manuscript technically sound, and do the data support the conclusions?

Reviewer #1: Yes

Reviewer #2: Yes

2. Has the statistical analysis been performed appropriately and rigorously? 

Reviewer #1: I Don't Know

Reviewer #2: Yes

3. Have the authors made all data underlying the findings in their manuscript fully available?

Reviewer #1: Yes

Reviewer #2: Yes

4. Is the manuscript presented in an intelligible fashion and written in standard English?

Reviewer #1: Yes

Reviewer #2: Yes

5. Review Comments to the Author

Reviewer #1: Dear author,

Overall is an interesting article in a very relevant topic for which there are substantial knowledge gaps in the literature and therefore, field studies like this are needed. The paper provides a very powerful message about the "Clinical efficacy and safety of interferon (Type I and Type III) therapy in patients with COVID-19: A systematic review and meta-analysis of randomized controlled trials".

Reviewer #2: Really great work summarizing the evidence for interferon for COVID-19.

A few minor comments, most of which can be addressed by changing some of the body text.

1) In the background, I would specify which clinical outcomes as they vary so drastically between studies and should be described with a bit more detail (e.g., discharge, viral load, etc?)

For example here:

These IFNs have been reported to exert favorable effects for treating viral infections, such as

71 hepatitis B and C, as well as SARS and MERS [11-13]. Since the COVID-19 pandemic, IFNs have

72 been evaluated in several randomized controlled trials as attractive therapeutic candidates due to

73 their plausible mode of action, suppression of IFN activity in severe COVID-19, and inhibition of

74 SARS-CoV-2 replication in vivo and in vitro [14]. Seminal studies demonstrated that systemic IFN-

75 β resulted in clinical benefits in hospitalized patients with mild-to-moderate COVID-19 [15, 16].

76 However, a large randomized controlled trial (RCT) did not reveal additional benefits of the

77 combination therapy of IFN-β and remdesivir compared to remdesivir alone in hospitalized

78 patients with moderate-to-severe COVID-19 [17].

2) I understand why the last review was in March 2022 (as stated below) but I would be curious if its worth checking for updated articles now given its been nearly a year? Or acknowledging this limitation more directly.

Since

99 new evidence on treatments for COVID-19 is continuously produced, the search was updated on the

100 10th day of each month, starting from August 2021 to March 2022.

3) I appreciated the use of standardized definitions of covid-19 severity. This is a strength of the study.

4) Can you include more details on what the serious adverse events were? Definitions may vary across studies.

5) For this comment, Although two RCTs on

350 mild COVID-19 were included, the primary outcome in these studies was microbiologic

351 outcome that assessed time to viral negativity or proportion of viral negativity at day 7 based

352 on SARS-CoV-2 PCR [27, 28].

I do agree with this point but I do think that viral negativity may be worth considering given spread of disease/ viral shedding/ etc. If interferon does reduce these, that would be a good finding in light of vaccines not stopping infections just reducing symptoms/severity. I don't recommend you re-do the analysis, but rather provide a sentence of two to highlight this.

6) A couple of methodological points - how heterogenous were these studies in terms of the study outcomes and in the control arms? I see one comment in the limitations and I see the column in Table 1 but some are unclear to me such as 'time to NEWS2<2" and "decline to two steps on the seven step ordinal scale". Can you make these outcomes a bit clearer in the table?

You may want to comment on this as a limitation of the meta-analysis or elaborate further in the discussion - a sentence should be fine.

7) Population - I see you included adults.. were there any other specifics about the populations? For example, pre-existing conditions? It may be good to report this in Table 1 to fully describe the studies.

6. PLOS authors have the option to publish the peer review history of their article (what does this mean?). If published, this will include your full peer review and any attached files.

Reviewer #1: No

Reviewer #2: No

---

## [Author Response · Author response to Decision Letter 0]

27 Feb 2023

Detailed Response to Reviewers

Notes: The reviewers' comments are in boxes; and our responses are in plain text. 

Reviewer #1: 

Overall is an interesting article in a very relevant topic for which there are substantial knowledge gaps in the literature and therefore, field studies like this are needed. The paper provides a very powerful message about the "Clinical efficacy and safety of interferon (Type I and Type III) therapy in patients with COVID-19: A systematic review and meta-analysis of randomized controlled trials"

Authors' response to comment #1>

We really appreciate for your critical review and comment. We wish that this meta-analysis for interferon as an antiviral agent can contribute to help clinical trial for COVID-19 or other emerging infectious diseases in future. 

Reviewer #2: 

Really great work summarizing the evidence for interferon for COVID-19.

A few minor comments, most of which can be addressed by changing some of the body text.

 

Comment #1

Authors' response to comment #1>

We completely agree with your comment. Although we did not recognize it while drafting, this paragraph was not clearly written. The paragraph you mentioned has been revised as follows

“These IFNs have been reported to exert favorable effects such as viral suppression in hepatitis B and C as well as lower mortality and faster improvement of chest radiograph in SARS and MERS [11-13]. Since the COVID-19 pandemic, IFNs have been evaluated in several randomized controlled trials as attractive therapeutic candidates due to their plausible mode of action, suppression of IFN activity in severe COVID-19, and inhibition of SARS-CoV-2 replication in vivo and in vitro [14]. Seminal studies demonstrated that systemic IFN-β was more likely to alleviate symptoms and shorten the duration of viral shedding in hospitalized patients with mild-to-moderate COVID-19 [15, 16]. However, a large randomized controlled trial (RCT) did not demonstrate that the combination therapy of IFN-β and remdesivir can lead to reduce the time to clinical recovery, compared to remdesivir alone in hospitalized patients with moderate-to-severe COVID-19 [17].

Comment #2

Authors' response to comment #2> 

 Thank you for the comment. And yes, we have updated as of January 23 of 2023 to follow up on the most recent articles. However, there were no articles that met the PICO and therefore the results have not been changed. Search Strategy and Figure 1. PRISMA flow chart has been revised.

On the manuscript the revised description is as follows.

A total of 8,305 articles were retrieved from the databases on 11 June 2021. After excluding duplicates, 7,170 articles were identified. After the living search update, total of 7,826 records were screened as of 23 January 2023. Based on the selection criteria, 137 articles were selected for full-text review.

 

Comment #3

Authors' response to comment #3>

Thank you for the appreciation. Yes, we believe it is one of our strengths on the paper which can directly benefit as a guideline for practitioners.

Comment #4

Authors' response to comment #4>

Thank you for your kind comments. Generally, serious adverse events in clinical trials are defined as 1) death or life-threatening event, 2) hospitalization (initial or prolonged), 3) Disability or permanent damage, 4) Congenital anomaly or birth defect, 5) Required intervention to prevent permanent impairment, 6) Other serious medical events such as anaphylaxis, seizure or emergency room visit. In this meta-analysis, some studies did not describe serious adverse events in detail. Based on generally accepted definition, we defined serious adverse events as follows in the section of data synthesis and statistical analysis

“As for serious adverse events, we defined it as generally accepted definition: 1) death or life-threatening event, 2) hospitalization (initial or prolonged), 3) Disability or permanent damage, 4) Congenital anomaly or birth defect, 5) Required intervention to prevent permanent impairment, 6) Other serious medical events such as anaphylaxis, seizure or emergency room visit.”

 

Comment #5

Authors' response to comment #5>

We absolutely agree with your opinion. It is important to assess microbiological outcomes such as viral negativity or viral load in mild COVID-19 when evaluating effectiveness of antiviral agents. We think that it is required to evaluate if IFN therapy as an antiviral agent can lead to reduce viral load faster, compared to placebo. Thus, we have added the following sentences in the limitation section.

“Feld et al showed that IFN therapy can significantly reduce duration of viral shedding, whereas the study of Jagannathan did not demonstrate it. It means that lowering viral shedding can have potential to prevent clinical worsening. Thus, future clinical trial needs to demonstrate whether IFN therapy can reduce viral load in patients with mild COVID-19.”

 

Comment #6

Authors' response to comment #6

Thank you for your comments. Study outcomes for clinical improvement were heterogeneously described among studies. Some studies defined clinical improvement as NEWS2 <2, and other used clinical improvement as improvement of ≥ 2 grade on predefined scale. Both NEWS2 <2 and improvement of ≥ 2 grade on predefined score or scale mean clinical improvement. We did not consider that different description for similar clinical outcomes can be difficult for reader to interpret it. Thus, we revised the study outcomes presented in table 1 to make them clearer using the term “clinical improvement”.

Comment #7

Authors' response to comment #7

Thank you for your kind comments. We have tried to add underlying conditions in table 1 as much as possible. However, some studies included patients with different severity group like moderate and severe, and underlying conditions were not presented for different severity group. In this case, we roughly described about underlying conditions using table footnotes. Other studies did not provide the information for comorbidity.

---

## [Decision Letter · Decision Letter 1]

8 Mar 2023

Clinical efficacy and safety of interferon (Type I and Type III) therapy in patients with COVID-19: A systematic review and meta-analysis of randomized controlled trials

PONE-D-22-21005R1

Dear Dr. Heo,

We’re pleased to inform you that your manuscript has been judged scientifically suitable for publication and will be formally accepted for publication once it meets all outstanding technical requirements.

Kind regards,

Ricardo Ney Oliveira Cobucci, Ph.D

Academic Editor

PLOS ONE

Additional Editor Comments (optional):

Reviewers' comments:

Reviewer's Responses to Questions

**Comments to the Author**

1. If the authors have adequately addressed your comments raised in a previous round of review and you feel that this manuscript is now acceptable for publication, you may indicate that here to bypass the “Comments to the Author” section, enter your conflict of interest statement in the “Confidential to Editor” section, and submit your "Accept" recommendation.

Reviewer #2: All comments have been addressed

2. Is the manuscript technically sound, and do the data support the conclusions?

Reviewer #2: Yes

3. Has the statistical analysis been performed appropriately and rigorously? 

Reviewer #2: Yes

4. Have the authors made all data underlying the findings in their manuscript fully available?

Reviewer #2: Yes

5. Is the manuscript presented in an intelligible fashion and written in standard English?

Reviewer #2: Yes

6. Review Comments to the Author

Reviewer #2: Great job addressing all of my prior comments. This research makes an important contribution to the literature and outlines gaps in understanding for future work.

7. PLOS authors have the option to publish the peer review history of their article (what does this mean?). If published, this will include your full peer review and any attached files.

Reviewer #2: No

---

## [Editor Report · Acceptance letter]

17 Mar 2023

PONE-D-22-21005R1 

Clinical efficacy and safety of interferon (Type I and Type III) therapy in patients with COVID-19: A systematic review and meta-analysis of randomized controlled trials 

Dear Dr. Heo:

I'm pleased to inform you that your manuscript has been deemed suitable for publication in PLOS ONE. Congratulations! Your manuscript is now with our production department. 

Kind regards, 

on behalf of

Dr. Ricardo Ney Oliveira Cobucci 

Academic Editor

PLOS ONE